# A Novel Packaging of the MEMS Gas Sensors Used for Harsh Outdoor and Human Exhale Sampling Applications

**DOI:** 10.3390/s23115087

**Published:** 2023-05-26

**Authors:** Lungtai Chen, Chinsheng Chang, Liangju Chien, Borshiun Lee, Wenlo Shieh

**Affiliations:** 1Smart Sensing and Systems Technology Center, Industrial Technology & Research Institute, Tainan 70955, Taiwan; jimmychang@itri.org.tw (C.C.); liangju@itri.org.tw (L.C.); borshiun@itri.org.tw (B.L.); 2Avantpac Technology Corporation, Kaohsiung 80673, Taiwan; wlshieh@avantpac.com

**Keywords:** MEMS gas sensors, COVID-19, human exhale sampling, harsh outdoor, packaging, PTFE filter, self-anchoring, transfer molding

## Abstract

Dust or condensed water present in harsh outdoor or high-humidity human breath samples are one of the key sources that cause false detection in Micro Electro-Mechanical System (MEMS) gas sensors. This paper proposes a novel packaging mechanism for MEMS gas sensors that utilizes a self-anchoring mechanism to embed a hydrophobic polytetrafluoroethylene (PTFE) filter into the upper cover of the gas sensor packaging. This approach is distinct from the current method of external pasting. The proposed packaging mechanism is successfully demonstrated in this study. The test results indicate that the innovative packaging with the PTFE filter reduced the average response value of the sensor to the humidity range of 75~95% RH by 60.6% compared to the packaging without the PTFE filter. Additionally, the packaging passed the High-Accelerated Temperature and Humidity Stress (HAST) reliability test. With a similar sensing mechanism, the proposed packaging embedded with a PTFE filter can be further employed for the application of exhalation-related, such as coronavirus disease 2019 (COVID-19), breath screening.

## 1. Introduction

Due to the rapid development of the Internet of Things (IoT) [1,2], there is an increasing demand for gas sensors used for indoor [3,4] or outdoor [5,6,7,8,9] air quality monitoring. Airborne dust particles typically carry a certain degree of charge or other attached substances. Moreover, high-humidity environments often contain condensed water vapor or mist. These two factors can vary significantly in harsh environments, leading to momentary errors or abnormal sensing in MEMS gas-sensing components. When dust or condensed moisture comes into contact with the sensing material of MEMS gas sensors, the sensing resistance changes, causing a relative change in value and leading to momentary errors or abnormal sensing in gas sensing.

Finding a non-invasive, painless, and simple approach to monitor health parameters and early detection of physiological disorders is of utmost importance in today’s world. Breath analysis has been considered a potentially powerful tool for studying medical diagnosis diseases due to its non-invasive nature and real-time monitoring capability [10,11,12]. It is used to detect diseases such as cancers [13,14,15,16] or diabetes [17,18], monitor disease progression, and track physiological data such as diet and exercise [19,20]. Alcohol breath tests are also one of the popular applications for human exhaled sampling [21,22,23,24]. The pre-driving alcohol test for transportation vehicle drivers is particularly helpful in reducing traffic accidents.

COVID-19 has spread all over the world, causing economic and health problems in various countries. Many research institutes have begun to research and develop an expiratory rapid screening method [25,26,27,28,29]. Like the aforementioned disease diagnosis and alcohol, it uses a gas sensor to detect the presence of specific gas compositions or viruses in the exhaled sample of the human body. However, high-humidity breath samples may contain or be accompanied by a certain degree of fine saliva, which is a possible source of misjudgment in gas sensing.

MEMS gas sensors are widely used in applications such as air quality monitoring, driving alcohol testing, health monitoring, and disease prevention due to their low power consumption and cost-effectiveness. Metal-oxide (MOX) gas sensors have diverse applications but share similar operational principles and architecture, consisting typically of a substrate, a heater, and a gas-sensitive layer [30]. Various manufacturing technologies, including Micro Electro-Mechanical System (MEMS) [31,32,33,34,35,36] and Complementary Metal Oxide Semiconductor (CMOS) [37,38,39,40], are used. In MEMS gas-sensing elements, the sensing material is typically heated to a high-temperature state during sensing operation. Due to the small thermal mass of the sensing material, the temperature is very sensitive to the gas samples being analyzed [41,42,43]. If any tiny dust particle or condensed water vapor, such as saliva, falls or attaches to the high-temperature sensing material during gas detection, the temperature change of the sensing material can shift the output value of the gas sensing concentration. This can lead to misjudgment and sensing offset in gas sensing. PTFE is a highly hydrophobic, water-repellent thermoplastic with excellent thermal stability and chemical resistance properties [44]. It is a crystalline polymer with a high melting temperature of approximately 342 °C [45] and can operate within a wide range of temperatures, from −190 °C to 260 °C [46]. PTFE film is an ideal filter material for high-temperature dust filtration industries, such as oil–water separation [47], liquid–air separation [48], and removal of ultrafine particles in PM2.5 [49]. Moreover, PTFE film has been adapted to improve selectivity characterization of MEMS gas sensors [50] and served as a sensing substrate of flexible humidity sensors [51]. It has also been employed by many MEMS gas sensor players, such as Sensirion AG [52,53] and Figaro Engineering Inc. [54], among others.

PTFE film is commonly used as an additional packaging part in gas sensor packaging and is typically attached to the sensor packaging by using an adhesive approach. However, the hydrophobic characteristic of PTFE can adversely affect the binding strength between the adhesive layer and the PTFE film. Additionally, the extra attachment process increases the complexity and total cost of the packaging process. Moreover, the white color of PTFE film makes it unsuitable for direct laser marking due to low label contrast. To address these issues, this study proposes a novel packaging method for MEMS gas sensors by applying a self-anchoring mechanism to embed a hydrophobic PTFE filter into the upper cover of the gas sensor packaging. In this approach, the PTFE film is unified into a functional packaging part, i.e., the upper cover, rather than an additional part.

## 2. Design and Manufacturing

### 2.1. Packaging Design

Loading a PTFE filter layer with hydrophobic properties is the most direct way to prevent the MEMS gas sensors from the influence of the gas sampling process. The filter layer’s function is to prevent dust particles or saliva samples from entering the air intake or exhaust holes of the gas sensing package. However, odor samples can still penetrate the filter layer and enter inside the sensor package. This approach enables the gas sensor to yield real sensing results without the interference of environmental dust or saliva from exhaled sampling or harsh outdoor conditions. In general, a couple of openings are designed on a gas sensor package to play the role of sensing entrance of the air or odor samples. To address issues related to packaging robustness and automated packaging, an upper cover embedded with the PTFE filter layer is designed in this study. Figure 1 depicts the schematic diagrams of the sensing situations of the MEMS gas sensor packaging with and without an embedded PTFE filter.

PTFE is a material with high hydrophobicity and vertical porosity that is often used for filtering or lubricating functions. The PTFE membrane, model 2501LT-WOPN240-G3, used in this study was obtained from EF-Materials Industries Inc., Taoyuan, Taiwan, and prepared through an extrusion and stretching process. The PTFE membrane comprises solid skeletons and filamentary porous spaces. The filamentary porous space runs through the entire PTFE membrane in an irregular shape and is responsible for the penetration channels of sensing air samples in this design. However, the hydrophobic characteristic of PTFE can adversely affect the binding strength between the adhesive layer and the PTFE film when used as an additional packaging part, as discussed in the previous section.

To solve the poor adhesion issue of the PTFE membrane, this study proposes an innovative packaging structure for MEMS gas sensors. The PTFE filter is stably and automatically loaded into the package body by using a common transfer molding process. The design principle is to anchor or seal the PTFE filter together with the upper cover of the sensor package using the molding pressure of the transfer molding process. The liquefied high-temperature packaging material of the upper cover, epoxy molding compound (EMC), is pressed to infiltrate into the partial porous space of the PTFE filter layer body. After the mold cools down, the solidified EMC press-infiltrated into the porous space of the PTFE filter layer naturally forms an anchoring composite layer structure between the EMC material of the upper cover and the PTFE filter layer material. This anchoring composite layer can firmly bond the PTFE filter layer to the upper cover, resulting in the PTFE filter layer being embedded in the upper cover. The PTFE filter prevents environmental dust or saliva from entering the package body, ultimately avoiding sensing interference.

### 2.2. Manufacturing

A PTFE film is a flexible material and cannot be directly placed on a heated mold. The PTFE film may become distorted due to thermal stress. To fix the PTFE film in position, a molding frame made of enforced circuit board with a thickness of 0.1 mm is used to solve this issue. The PTFE film is then adhered to the molding frame using a thermoset adhesive material before being placed onto the heated mold.

To ensure complete filling of the mold space, the surface of the PTFE film is designed with a punching pattern of crossing holes. High-temperature liquefied EMC flows through the crossing holes of the PTFE film and fills up all the inner empty spaces of the clamped mold during a transfer molding process. Figure 2 shows images of the PTFE film with the punching pattern of cross holes used in this study. EMC is usually composed of fillers and epoxies. The completeness of the molding filling is usually influenced by the size distributions of the filler. Two types of filler size distributions, single distribution and two-distribution, are employed in this study. The particle size distributions of the molding filler considered in this study are 20 µm, 5 µm + 15 µm, and 5 µm + 40 µm. After quality checking the molded upper cover, the molding filter of 5 µm + 15 µm particle distribution is selected as the final material composition of the EMC for the proposed gas sensor packaging.

Transfer molding is an automated packaging process employed in this study to form the upper cover embedded with a PTFE filter. Figure 3 illustrates the transfer molding process of the upper cover embedded with a PTFE filter. First, the lower mold was preheated to an operational temperature of around 180 degrees Celsius. Then, the molding frame, which was adhered with the PTFE film, was placed on the lower mold using a specific alignment mechanism. The upper mold was then lowered to complete the mold clamping process. An EMC plate with a specific filler recipe was heated to liquefy and squeezed into the clamped mold. The liquefied EMC materials flowed through the crossing holes of the PTFE film and filled up the inner space of the mold under a molding pressure of 450 psi. After maintaining the molding pressure for more than 20 s, the upper mold was raised, and the molded packaging part was finally ejected from the mold.

## 3. Tests

In order to confirm the packaging performance of the proposed packaging mechanism, two evaluation tests, high-moisture response and HAST, are estimated in this study.

### 3.1. High-Moisture Response Test

Sensing errors of the gas sensor can be attributed to various factors, such as the presence of condensed water vapor in high-humidity ambient air or saliva in high-humidity human breath samples. The purpose of the high-moisture response test is to verify the ability of the proposed innovative MEMS gas sensor packaging to retard the effects of moisture.

In this study, two types of packaging were used to explore the performance of the PTFE filter: one with a PTFE filter and the other without. The first type was a normal packaging without an embedded PTFE filter underneath the intake and vent holes, while the second type was the proposed packaging, with an embedded PTFE filter. MEMS alcohol sensors were packaged in both types, and both underwent a burn-in process to stabilize the sensing quality of the alcohol sensor chips. The burn-in condition involved keeping the burn-in current at 1.4 V for 88 h. A total of 10 testing specimens for both packaging types were used in this evaluation.

For testing and verification purposes, the 10 test specimens were placed in an 8-site evaluation socket, and an Alcohol Breath Simulator (model Guth 12V500, Guth Laboratories Inc., Harrisburg, PA, USA) was used to generate a specific test condition, which included a temperature of 34 °C, a specific humidity of 95% RH, and flow rate of 20 L/min. In general, an exhaled breath gas is greatly rich in moisture of 80–95% RH [55]. Three humidity conditions, 75% RH, 85% RH, and 95% RH, were incorporated to evaluate the filtration performance of the device. The moisture testing air was blown into a sealed testing box made of clear polymethylmethacrylate and kept for 25 s before being sent out the testing box. During this testing period of 25 s, a stabilized resistance, R_0_, and a reaction resistance of the gas sensor, R_a_, were recorded. The value of R_0_–R_a_ represents the sensing response of the gas sensor, which is proportional to the reaction level in response to the moisture testing conditions. The sensing response difference between the test specimen types, with and without the embedded PTFE filter, was defined as the retardation value. A measure of the retardation rate was used to estimate the function of the embedded PTFE filter in maintaining the stability of the moisture effect. Figure 4 shows the measurement setup of the high-moisture response test and a close-up view of the test specimen setup.

### 3.2. HAST Test

Sensor packaging embedded with a PTFE filter were subjected to a HAST test to verify the reliability of the proposed packaging. The HAST test was performed following an international testing standard, the EIA/JEDEC standard method A110-B. According to the normal testing specifications, a sample size of 25 and testing conditions of 130 °C/85% RH/96 h were set for the HAST test. Two performance parameters, response and sensitivity, of the gas sensor were investigated to evaluate the proposed gas sensor packaging in this study. Response refers to the change of the sensor’s output resistance in the presence of a specific alcohol concentration. The statutory limit of breath alcohol concentration (BrAC) for a traffic vehicle driver in Taiwan is 0.15 mg/L. Two alcohol concentrations, 0.1 mg/L and 0.2 mg/L, were utilized in this study. By converting the alcohol molecule, the corresponding alcohol concentrations used in this study are 52 ppm and 104 ppm BrAC, for 0.1 mg/L and 0.2 mg/L, respectively. Sensitivity represents the rate of response change per unit alcohol gas concentration, and its unit is 1/ppm. To assess the sensitivity characterization of the proposed gas sensor packaging, a concentration range between 52 ppm and 104 ppm was established before and after undergoing the HAST test. Regarding the evaluation procedure of the HAST examination, we recorded the two aforementioned performance parameters for a total of 25 test specimens in advance. Subsequently, the 25 test specimens were placed inside the testing chamber of the HAST instruction to initiate the formal HAST test. The same two performance parameters were measured for the 25 test specimens to enable further data comparison. Ultimately, a HAST test report was generated in accordance with the failure criterion listed in the international testing standard, EIA/JEDEC standard method A110-B.

## 4. Results and Discussion

Using the transfer molding process to embed a PTFE filter in the upper cover is the innovative approach proposed in this study. Figure 5 shows an SEM diagram of a cross-section of the packaging embedded with a PTFE filter and a top-view image of the upper cover. Figure 6 shows cross-section SEM diagrams of the self-anchoring mechanism. The PTFE filter is completely embedded in the correct position of the upper cover, located below the intake and vent holes of the upper cover. The purpose of the PTFE filter is to prevent environmental dust or saliva from interfering with the gas sensor chip during the sampling process.

A self-anchoring mechanism was created on the interface between the EMC material of the upper cover and the PTFE filter layer. During the transfer molding process, some of the liquefied epoxy and tiny solid particles of the EMC were forced and infiltrated into the spaces of the filamentary porous PTFE filter by the molding pressure. The cooled-down EMC of the upper cover then occupied the filamentary porous space partially and joined the EMC with the PTFE filter layer tightly.

The crown-shaped materials shown in Figure 6 are the original solid skeleton of the PTFE filter and are all located at the interface between the PTFE filter and the EMC material of the upper cover. The self-anchoring layer is composed of the solid EMC material that has partially filled the filamentary porous space of the PTFE filter and the crown-shaped material. The depth of the self-anchoring mechanism layer depends on two primary parameters, namely the molding pressure parameter of the transfer molding process and the inner shape of the filamentary porous space of the PTFE filter.

Air is a compressible medium in nature. The more compression applied to air sealed in a closed space, the smaller the air space volume will be. Generally, the greater the molding pressure that is set, the deeper the penetrated depth of the liquefied EMC material is reached. In other words, if a higher molding pressure is set, a thicker self-anchoring layer will be formed. Sixty-two measurements of the self-anchoring layer thickness were collected, and a mean thickness of 23.1 µm with a standard deviation of 5.8 µm was recorded. The self-anchoring layer can be found on both interfaces of the PTFE filter layer. As a result, the total thickness of the self-anchoring layer is 46.2 µm, which is roughly 18% of the total PTFE filter. The self-anchoring layer only exists at all of the contact interfaces between EMC and the PTFE filter materials. The PTFE filter is eventually embedded in the upper cover, to be a part of the upper cover. The self-anchoring layer binds the PTFE filter layer and upper cover tightly, and no delamination or crack is observed at the interface of the self-anchoring layer.

The filamentary porous spaces in the PTFE membrane are normally present in an inter-closing format. In other words, the air-penetrating channels in the PTFE membrane are interconnected. The whole volume of closed filamentary porous spaces was reduced when the liquefied EMC was pushed into the filamentary porous spaces by the molding pressure during the transfer molding process. As a result, the entire closed filamentary porous spaces presented a positive inner pressure when the intruded EMC solidified. However, the positive inner pressure sealed in the closed filamentary porous spaces will be eventually released following the mold-releasing process. Since the whole filamentary porous spaces in the PTFE membrane are interconnected, all the air contained in the closed filamentary porous spaces will reach an equalization status with the outside atmosphere pressure by passing through the filamentary pores located at the intake or vent holes position of the upper cover. Therefore, the inner pressure will be released naturally, and the packaging will be stress-free. The stress-free issue of packaging embedded with a PTFE filter will be verified through the further HAST test.

Regarding the PTFE filtering layer situated just underneath the intake and vent holes of the upper cover, the gas samplings have to penetrate the PTFE filter layer and react with the sensing material of the gas sensor chip during gas sensing operation. The embedded PTFE filter must have a smooth penetration tunnel across the whole thickness direction of the PTFE filter layer in the upper cover. The SEM cross-section diagram of the whole thickness direction and a detail view of the PTFE filter are shown in Figure 7. Liquid cold-mount epoxy is a medium used to enclose a cross-section specimen during the sample preparation process of the cross-section inspection. It is not an original part of the cross-section specimen. None of the cold-mounted epoxy material shown in Figure 7 is detected in the filamentary porous space of the PTFE filter. Due to the essential hydrophobic characteristics of the PTFE material, none of the liquid materials can penetrate into the filamentary porous space of the PTFE filter, even for a binder adhesive material. This is the key reason why PTFE film cannot be bound or attached to any surface strongly by any adhesives, but the self-anchoring mechanism proposed in this study can.

SEM diagrams of top view of the upper cover with a whole view and cross-up view of the underneath PTFE filter are shown in Figure 8. In order to ensure that the channels in the PTFE filter underneath the intake and vent holes remain unobstructed, the original surface of the PTFE filter in these areas must be maintained. Figure 8 demonstrates that the filamentary porous space and solid skeleton distribution patterns are clearly visible over nearly the entire surface area of the PTFE filter in the intake and vent holes of the upper cover, which indicates the success of the self-anchoring mechanism and confirms that all gas samples can penetrate into the sensing chamber of the gas sensor. However, a small amount of EMC overflow was also observed on the border of the intake and vent holes, which could be improved by optimizing the molding clamping pressure and mold interference design between the upper and lower molds during the mass-production optimization program.

Two types of tests, high-moisture response and HAST, were conducted to verify the packaging performance proposed in this study. Regarding the high-moisture response test, the changes in sensing resistance (R_0_–R_a_) for 10 gas sensor specimens, both with and without a PTFE filter, are presented in tabular form. Figure 9 provides the collected data for the high-moisture response tests conducted at 75% RH, 85% RH, and 95% RH, respectively. A higher change in sensing resistance of the packaged gas sensor indicates higher moisture sensitivity. The retarding rate is an index used to estimate the moisture sensitivity level for both packaged gas sensor specimens. A positive retardation rate means that the humidity sensitivity for packaging specimens without a PTFE filter is lower than that with a PTFE filter. Figure 9 illustrates the prevention performance of the gas sensor packaging embedded with a PTFE filter under three different humidity conditions, 75%, 85%, and 95% RH. Typically, humidity causes a resistance drop in an n-type MOX gas sensor and exhibits a saturation reaction under high humidity conditions [56]. Both the gas sensor with and without the embedded PTFE filter exhibit the same trend in this study. The gas sensor packaged with the PTFE filter demonstrates lower moisture sensitivity and resistance drop compared to the one without the PTFE filter across all three humidity conditions. All the packaging specimens embedded with the PTFE filter exhibit a positive retardation rate in the three humidity conditions. The mean retardation rates of the packaged gas sensor embedded with the PTFE filter are 60.6%, 59.9%, and 57.3% for the humidity conditions of 75% RH, 85% RH, and 95% RH, respectively. The slight variation in retardation rates among the three humidity conditions could be attributed to a stabilized characterization of the gas sensor. The gas sensor embedded with the PTFE filter exhibits a lesser sensitivity to humidity compared to the gas sensor without a PTFE filter. However, the aforementioned positive retardation rates demonstrate the preventive performance of the PTFE filter.

Figure 10 illustrates the recorded response performance of the 25 test specimens before and after undergoing the HAST test. The response performance of the 25 test specimens showed a slight degradation after the HAST test. HAST is essentially an accelerated stress test that applies stress to the micro-structure of the gas sensor chip, resulting in mechanical performance degradation [57]. However, all 25 test specimens remained functional and could still operate under the specific alcohol concentrations of 0.1 mg/L and 0.2 mg/L.

Figure 11 depicts the sensitivity performance of the 25 test specimens before and after the HAST test. Most of the 25 test specimens exhibited higher sensitivity performance within the alcohol concentration range of 0.1 mg/L and 0.2 mg/L after undergoing the HAST test. This can be attributed to the different degradation rates in response performance between the two alcohol concentrations for the 25 test specimens. Figure 9 indicates that the response degradation level was slightly higher for the test specimen soaked in 0.1 mg/L alcohol concentration compared to the one soaked in 0.2 mg/L alcohol concentration. Generally, a higher sensing concentration situation approaches the sensing saturation condition of a gas sensor more closely compared to a lower sensing concentration. Consequently, a gas sensor operated in a concentration approaching its saturation point will exhibit lower sensitivity characterization. In other words, any external influences coming from the environment will have minimal impact.

Regarding the reliability evaluation of the HAST test for packaged gas sensors embedded with a PTFE filter, all 25 specimens exhibited normal functional output after undergoing the HAST test. Based on the failure criterion specified in EIA/JEDEC standard Test Method A110-B document, the proposed novel packaging specimens have passed the HAST test.

## 5. Conclusions

This study presents a novel packaging approach for MEMS gas sensors. A successful self-anchoring mechanism is demonstrated to bind together the upper cover and PTFE filter. The PTFE filter located just underneath the intake and vent holes of the upper cover effectively prevents the sensing material of the gas sensor chip from interference by environmental dust and saliva exhaled from human breath samples. The proposed packaging passed both HAST and high-moisture response tests. Additionally, the proposed packaging adopts a widely used transfer molding process, which simplifies the packaging process and reduces packaging cost. By following the same packaging approach of the proposed novel mechanism, it will be possible to adapt the sensor packaging for exhale-type COVID-19 quick screening agents, due to their high-moisture human breath sampling requirements.

## Figures and Tables

**Figure 1 sensors-23-05087-f001:**
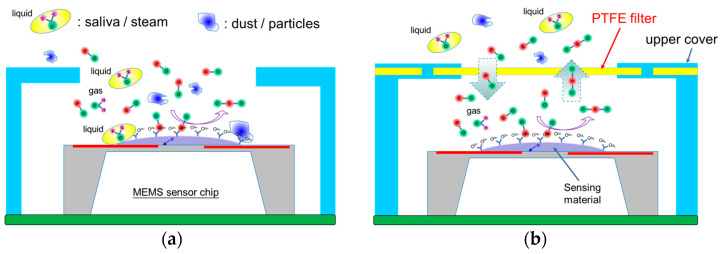
Schematic diagrams of sensing situations of the MEMS gas sensor packaging; (**a**) traditional packaging without a PTFE filter embedded and (**b**) proposed packaging with a PFTE filter embedded.

**Figure 2 sensors-23-05087-f002:**
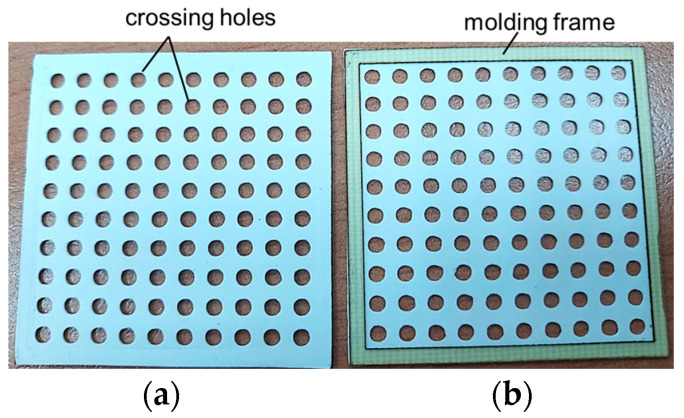
Images of the PTFE film used in this study, (**a**) with punched crossing holes and (**b**) fixed on a molding frame.

**Figure 3 sensors-23-05087-f003:**
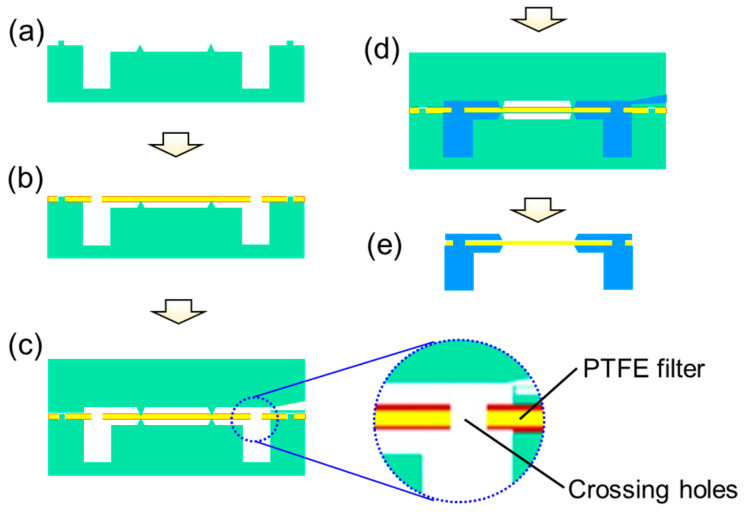
The schematic diagrams of transfer molding process for fabricating an upper cover embedded with a PTFE filter including (**a**) lower mold, (**b**) PTFE film setting, (**c**) mold closing, (**d**) liquefied molding compound squeezing, (**e**) mold releasing and molded parts ejected.

**Figure 4 sensors-23-05087-f004:**
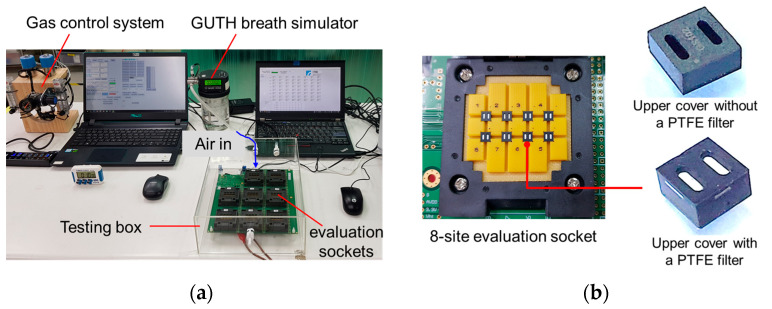
Measurement setup of the high-moisture response test; (**a**) whole view and (**b**) a close-up view of the test specimens setting.

**Figure 5 sensors-23-05087-f005:**
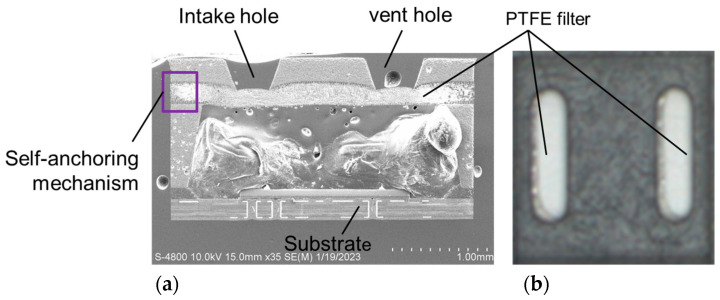
(**a**) SEM diagram of cross-section of the packaging embedded a PTFE filter and (**b**) top-view image of the upper cover.

**Figure 6 sensors-23-05087-f006:**
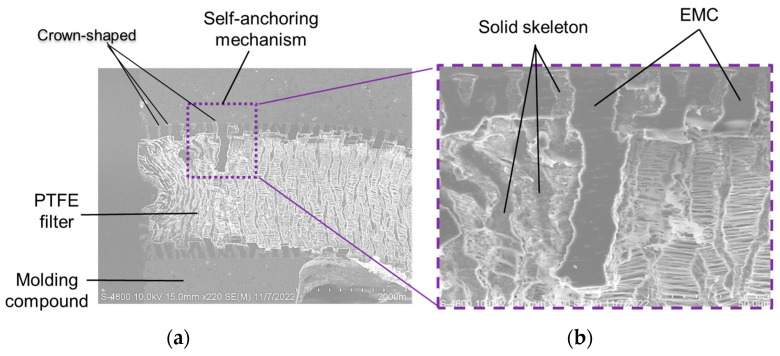
SEM diagrams of the self-anchoring mechanism for (**a**) a whole view and (**b**) its close-up view.

**Figure 7 sensors-23-05087-f007:**
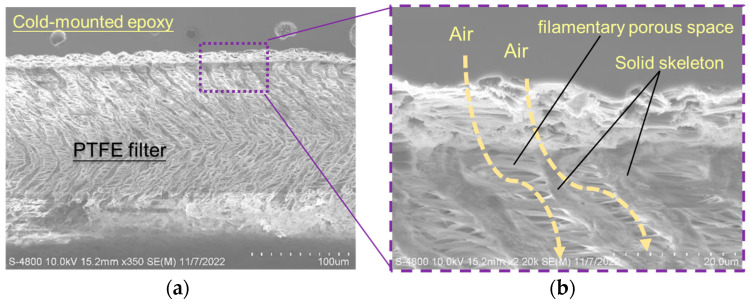
SEM crossing-section diagrams of (**a**) whole thickness direction and (**b**) detail view of the PTFE filter.

**Figure 8 sensors-23-05087-f008:**
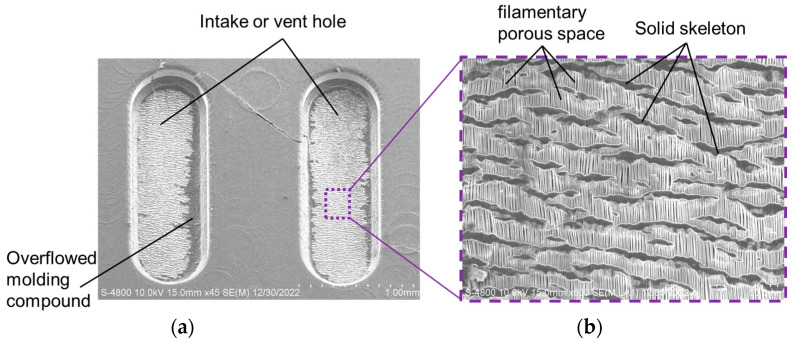
SEM diagrams of top view of the upper cover with (**a**) a whole view and (**b**) cross-up view of the underneath PTFE filter.

**Figure 9 sensors-23-05087-f009:**
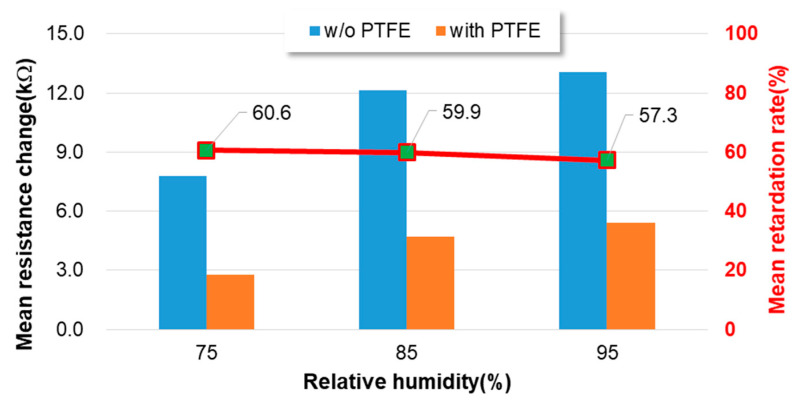
The prevention performance of the gas sensor packaging embedded with a PTFE filter under 75%, 85%, and 95% RH sensing conditions. The red line curve represents the average retardation rate of the gas sensor with the PTFE filter.

**Figure 10 sensors-23-05087-f010:**
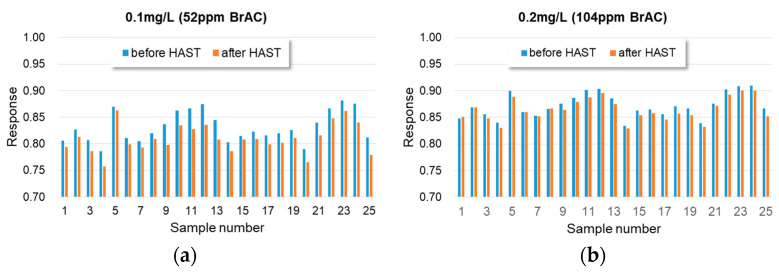
The recorded response performance of the 25 test specimens before and after experiencing HAST test, for (**a**) 0.1 mg/L alcohol concentration and (**b**) 0.2 mg/L alcohol concentration.

**Figure 11 sensors-23-05087-f011:**
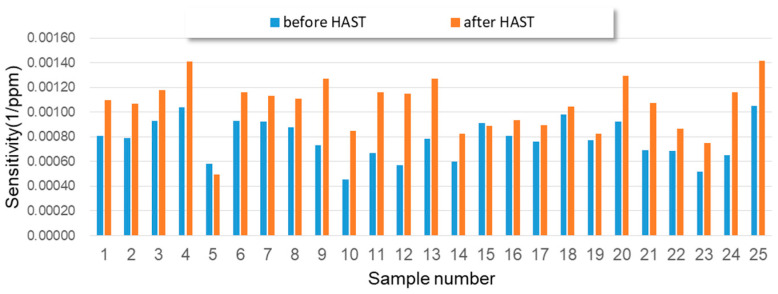
The sensitivity performance of the 25 test specimens before and after experiencing HAST test.

## Data Availability

Not applicable.

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
