# Peer review of "A Novel Packaging of the MEMS Gas Sensors Used for Harsh Outdoor and Human Exhale Sampling Applications"

_sensors, 2023, doi:10.3390/s23115087_

Round 1

Reviewer 1 Report

The presented paper " A novel packaging of the MEMS gas sensors used for harsh outdoor and human exhale sampling applications " proposes a novel packaging mechanism for MEMS gas sensors that utilizes a self-anchoring mechanism to embed a hydrophobic polytetrafluoroethylene (PTFE) filter into the upper cover of the gas sensor packaging. By this method, the influence of moisture or dust in the detection process of gas sensors is avoided, and the detection accuracy is improved. Overall, the manuscript is well-organized and written in an easy-to-understand manner. The data in the manuscript as well as the test results are also able to effectively support the ideas presented in the manuscript. However, this manuscript is not complete enough to be published in its current form. Therefore, after revising the following issues and modifications, I suggest that this manuscript be considered for sensors.

1.     There's an extra "T" on line 14 of the abstract on the first page of the manuscript.

2.     What advantages does PTFE have over other filtration materials? Why choose PTFE as the filter?

3.     In the manuscript, only the device performance was evaluated under a single humidity condition. What is the filtration performance of the device under different humidity conditions?

4.     The results obtained in this paper are mainly described in words. Although the method and principle of the proposed packing mechanism are clearly expounded, the experimental results are not intuitive enough. I suggest the author that the sensing performance and test data of the proposed gas sensor be displayed intuitively in the form of graphs rather than just text descriptions. For example, but not limited to the display of sensor response test results.

The English language needs to be further improved.

Author Response

Reply letter for reviewer 1 (Manuscript ID sensors-2406562)

The authors are deeply grateful to the reviewer for your recommendations in advance. The questions mentioned in the recommendation letter have been addressed in a point-by-point manner and are listed as follows:

Q1. There's an extra "T" on line 14 of the abstract on the first page of the manuscript.

A1: The extra letter “T” listed on line 14 has been erased.

Q2. What advantages does PTFE have over other filtration materials? Why choose PTFE as the filter?

A2: The reasons for selecting PTFE material as the filter layer in this study are listed below:

  • Hydrophobicity: PTFE prevents the entry of condensed water or saliva into the packaging body.
  • High-temperature tolerance: PTFE can withstand a wide range of temperatures, from -190°C to 260°C, making it compatible with the process temperature requirement of the transfer molding process. The process temperature of the transfer molding process is 180°C.
  • Further gas sensor investigation: A separate study is being conducted to enhance the sensing selectivity characterization of the gas sensor by applying a coating layer to the surface of PTFE.
  • Widely used: PTFE has already been utilized as a filter material in the packaging of several commercialized MEMS-based gas sensors, including those from Sensirion AG, a leading company in MOX gas sensor technology. For more details, please visit the following website address: https://sensirion.com/products/catalog/?category=VOC.

Q3. In the manuscript, only the device performance was evaluated under a single humidity condition. What is the filtration performance of the device under different humidity conditions?

A3: Two additional humidity testing conditions, 75% RH and 85% RH, were incorporated to evaluate the filtration performance of the device. The results demonstrate that the PTFE filter exhibited the anticipated trend of lower moisture sensitivity across all three humidity conditions: 75% RH, 85% RH, and 95% RH. The original 10 test samples had been utilized for another application as per the project arrangement. To ensure data consistency, we employed an additional set of 10 new specimens for the three additional humidity conditions. For more detailed information, please refer to page 5, 6, 9, and 10 of the revised manuscript file. The supplementary or revised content is highlighted in red letters.   

Q4. The results obtained in this paper are mainly described in words. Although the method and principle of the proposed packing mechanism are clearly expounded, the experimental results are not intuitive enough. I suggest the author that the sensing performance and test data of the proposed gas sensor be displayed intuitively in the form of graphs rather than just text descriptions. For example, but not limited to the display of sensor response test results.

A4: The sensing performance of the gas sensors, both before and after undergoing the HAST test, has been visually presented in the form of graphs. Two performance parameters of the gas sensors, response and sensitivity, were evaluated. The results for these two performance parameters of the gas sensors, even after undergoing the HAST test, are presented and show a reasonable performance. Please refer to page 6, 10, and 11 of the revised manuscript file for detailed information. The supplementary or revised content is highlighted in red letters.

  By the way, the revised manuscript has been reviewed by an alternative native speaker to enhance the readability of the submitted manuscript.

Reviewer 2 Report

The authors show a way to solve an urgent problem of gas sensors - to exclude the influence of dust and condensate water on the operation of the gas sensors in a real conditions, often hard conditions.

They suggest to embed a commonly used PTFE filter into the upper cover of the package using innovative self-anchoring mechanism instead of the current external pasting method. This new method looks promising and, according presented tests, can significantly improve the gas sensor performance in conditions of high-humidity and temperature as well as dust.

I recommend the article for publication and ask authors few questions.

  • Can this packaging method be used for all type of PTFE filters?
  • What upper cover (package lid) materials are suitable for this method?
  • Do the epoxy molding compound and the thermoset adhesive material you used outgas? How does this affect the performance of the sensor?
  • Is it realistic to convert the method to production?

Minor corrections

Author Response

Reply letter for reviewer 2 (Manuscript ID sensors-2406562)

The authors would like to express our sincere appreciation to reviewer for your recommendation of article publication in advance. The questions mentioned in the recommendation letter have been addressed in a point-by-point manner and are listed as follows:

Q1: Can this packaging method be used for all type of PTFE filters?

A1: Generally, the answer is yes. However, the porous size and structure of the PTFE film usually depend on the specific manufacturing process employed for film formation. The process yield of the proposed packaging method could be influenced by the two parameters mentioned above.

Q2: What upper cover (package lid) materials are suitable for this method?

A2: Two commonly used molding approaches, injection molding and transfer molding, can be employed to incorporate a PTFE film into a plastic package. Therefore, typical plastic materials used in the injection process and EMC (Epoxy Molding Compound) employed in the transfer process could be potential candidate materials for the package lid. Ensuring stress balance between the upper cover and the package body is a crucial design consideration when selecting the appropriate material. 

Q3: Do the epoxy molding compound and the thermoset adhesive material you used outgas? How does this affect the performance of the sensor?

A3: Regarding the concern of outgassing from the epoxy molding compound, several gas sensors in the market have been successfully packaged using this material. For example, companies like Sensirion AG, a leading player in MOX gas sensors, and AppliedSensor GmbH. The impact of outgassing from the epoxy molding compound on gas sensing has been effectively addressed. On the other hand, the thermoset adhesive material used between the PTFE film and the molding frame does not remain in the final sensor package. It is removed during panel trimming after the transfer molding process. Therefore, there are no outgassing issues associated with the gas sensor package proposed in this study.

Q4: Is it realistic to convert the method to production?

A4: Yes, a packaging player from Taiwan is collaborating with my organization in an effort to advance the implementation of this novel packaging concept.